# Nanoparticles from the Cosmetics and Medical Industries in Legal and Environmental Aspects

**Renata Włodarczyk *** and **Anna Kwarciak-Kozłowska**

Faculty of Infrastructure and Environment, Czestochowa University of Technology, Dabrowskiego 69,
42-201 Czestochowa, Poland; anna.kwarciak@pcz.pl
* Correspondence: renata.wlodarczyk@pcz.pl

**Abstract:** This paper presents the application and role of nanomaterials, with particular emphasis on the cosmetics and medical industries. Methods of obtaining materials at the nanoscale and their characteristic structure, which determines their attractiveness and risk, especially in recent years, have been described. The subject of the work was to indicate the hazards and risks that are associated with the properties of nanomaterials; dimension, and high chemical and physical activity, thus making ways to capture and monitor them difficult. Legal and environmental aspects were taken into account, and the involvement of the European Commission in this subject and the activities carried out in a few European countries as well as in Japan, the USA and Canada were analyzed.

**Keywords:** nanoparticles; nanomaterials; toxicity; ecotoxicity; safety; health

## 1. Application and Role of Nanoparticles in Cosmetology and Medicine

In October 2011, the European Union defined nanoparticles (NPs) as natural or artificially produced materials containing particles in unbound state or as aggregates (agglomerates) in which 50% or more of them occur in the size range from 1 nm to 100 nm [1–3]. Natural nanoparticles appear in the environment as a result of erosion, decomposition or oxidation of organic matter or minerals. A significant number of nanoparticles are released during forest fires or volcanic eruptions [4]. In the case of man-made nanoparticles, we are talking about those created unintentionally in various processes, including as by-products of combustion, mainly of diesel or wood (ultrafine fractions), welding, smelting or soldering, and similar products with intended properties, shape, size, called engineered inorganic nanoparticles (EINP) [5]. All known nanoparticles can be divided into two groups: organic nanoparticles (e.g., fullerenes and carbon nanotubes) and inorganic nanoparticles, which include metals (Ag, Au, Cu, Pa, Pt), metal oxides ($TiO_2$, $ZnO$, $Fe_2O_3$, $CuO$, $Fe_3O_4$), quantum dots (CgSe, CdTe) and sea salt [4,6–8]. Metallic nanoparticles are usually synthesized in two ways:

- the "top-down" method—in which the size of large structures is reduced to a nanometer scale by reducing the size (grinding) of materials to nanoparticles (products of the first generation);
- the "bottom-up" method—by building new structures based on nanoparticles, aggregation of molecules dissolved in the liquid or gas phase (second generation products) [9,10].

Among the "top-down" methods, the main role is played by physical processes (including mechanical/ball milling, thermal/laser ablation, sputtering, electro-explosion) as well as chemical etching. The "bottom-up" approach to nanoparticle synthesis involves chemical and biological methods. However, physical and chemical methods for the synthesis of nanoparticles are harmful to the environment, due to the use of high temperature, pressure, and hazardous chemicals. In addition, chemically synthesized nanoparticles can only be used in biomedical applications due to their smaller biocompatibility and

instability. Nanoparticles synthesized by bioreduction or using cell-free extract (supernatant or whole plant tissue/microbes) are characterized by less or no toxicity [9,11–13]. Biosynthesis of nanoparticles by microorganisms is classified as green and eco-friendly technology, and may be intracellular or extracellular according to the location of nanoparticles [14]. NPs are characterized by a specific geometric structure, distinguished by a high surface-to-volume ratio, which is greater the smaller the diameter of the particles. This is associated with an increase in the activity of the nanoparticle form, and it affects their absorption properties and reactivity. The shape and size of metal nanoparticles and their stability are affected by the method of their preparation, and the selection of appropriate surfactants ensures that nanoparticles of the desired shapes are obtained. By changing the molar ratio of the precursor and stabilizer, it is possible to obtain nanoparticles of various sizes [15]. In the presence of chemicals (including surfactants), the surface and interfacial properties of nanoparticles can be modified [16,17]. In general, NPs have significantly different physicochemical properties compared to small particles of the same composition [18]. Macroscopic silver melts at 960 °C, and its nanoparticles with a diameter of 2.4 nm melt at 360 °C. Nanoparticles can be both spherical, fibrous and layered. They appear as aerosols (mainly solid or liquid phase in the air), suspensions (mainly solid phase in liquids) or emulsions (two liquid phases). A very wide range of applications of nanoparticles in many technologies leads to the constant growth in products with their participation. Their number is evaluated at over 1000. It is estimated that their production in 2011–2020 will amount to about 58 thousand tons of nanomaterials [19]. Due to their specific physicochemical properties, nanoparticles are widely used in the chemical and food industries, in water and wastewater treatment, in photovoltaics or in electronic devices. NPs are also used as fuel additives (e.g., $CeO_2$) [19–23]. The mentioned directions of nanoparticles application are presented in Figure 1.

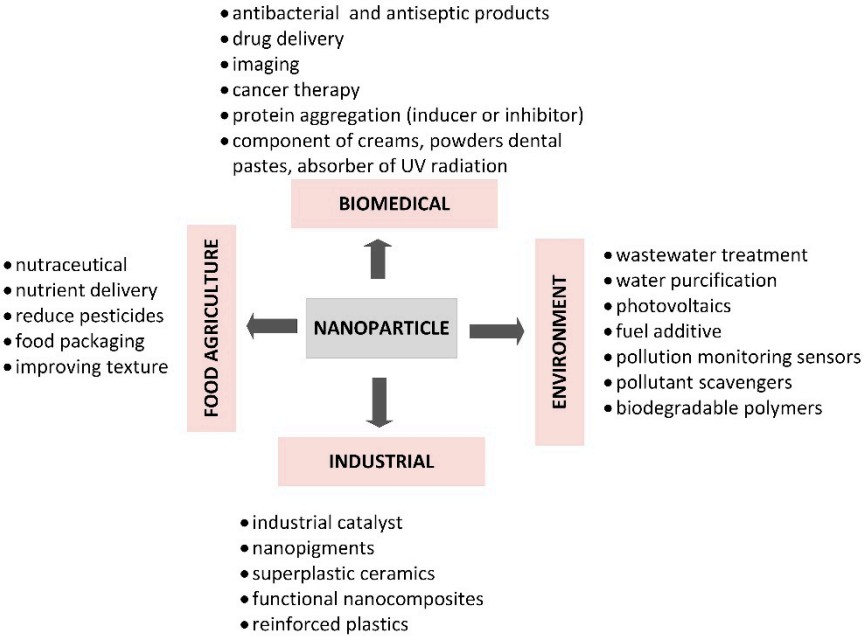

**Figure 1.** Directions using nanoparticles, modified from [2,20–24].

The development of nanotechnology has contributed to the emergence of alternative and competitive antibacterial and antiseptic agents, such as silver, gold and copper nanoparticles, which today appear to be free from the defects of traditional antibiotics [21,22]. The antibacterial properties of silver nanoparticles result from their small size and developed surface, which allow them to easily penetrate biological membranes and microorganisms, causing their death [24,25]. The action of nanoparticles on pathogenic organisms boils down to three main mechanisms:

- disruption of the electrical potentials of the cell membrane, nucleus and mitochondria (in bacteria);
- water management disorders (in the case of mushrooms);
- depriving the ability of catalytic degradation of the lipid-protein base (in viruses).

Silver and its bactericidal properties were already known in ancient Greece, where, among others, silver coins were used to purify water. Currently, silver nanoparticles can be found in fabrics (dressings, socks, underwear), cosmetics (powders, deodorants) and medicine (catheters, implants). The return to the use of silver as a bactericide in the form of solutions, suspensions or nanoparticle forms has its justification as it is deposited on various substrates, e.g., silica or polymeric, which are considered some of the most effective disinfectants, combining the biocidal and deodorizing properties of silver and silicon. Due to their inactivating properties of pathogens, silver nanoparticles are also included in containers for storing food or children's toys [26,27]. Copper nano preparation has strong anti-fungal activity. They can be added to products intended for oral hygiene to prevent inflammation. Gold in the nano form shows great ease of penetration into the cells and acts strongly in regenerating and stimulating [28].

In cosmetics, zinc oxide and titanium (IV) oxide are most commonly used. These types of compounds are components of so-called mineral cosmetics, which have been increasingly replacing traditional cosmetics in recent years. This is mainly because they are obtained from powdered, sterilized minerals that are mixed with pigments, processed without the use of chemical compounds. Cosmetics with nanoparticles are used in skin, hair, nail and mouth care [22,29]. The presence of NPs is, among other things, to reduce the effects of photoaging, wrinkles and discoloration, reduce hair loss and even prevent graving. Nanocosmetics are also designed to extend the fragrance's durability and cause its slower release. In sunscreen products, zinc oxide or titanium dioxide nanoparticles are considered the most effective minerals for protecting the skin from harmful radiation [30–33]. Zinc oxide is the only one that protects against both UVB (290–320 nm), UVA1 (340–400 nm) and UVA2 (320–340 nm) rays. Another advantage of using zinc oxide is its healing properties. It is part of the so-called Lassara paste used in skin diseases, such as juvenile acne, cold sores or cracked corners of the mouth. Zinc oxide is obtained by micronization, i.e., fragmentation to fine pollen, with a grain size of about 20 nm; in this form it is suspended in silicone powder. Thanks to modern processes, micronized ZnO is transparent, and the right size of particles to prevent it from entering the bloodstream. Zinc compounds are used in cosmetics only for external use [34–36]. Titanium dioxide, similar to zinc oxide, is a physical sunscreen, impervious to UVA and UVB rays. In its micronized form, it clouds the mixture and hinders the penetration of light, which makes it an effective UV radiation filter (even in cosmetics with high SPF factors). In addition, it is a weaker allergen than zinc oxide. However, when the titanium dioxide particle absorbs the quantum of light, it then becomes a semiconductor with mobile charges, which ultimately leads to the formation of three moles of the hydroxyl radical, i.e., the strongest pro-oxidant, which destroys cells and deepens the aging process. Therefore, paradoxically, cosmetics containing titanium dioxide accelerate the aging process, instead of inhibiting it. In cosmetics, titanium dioxide is commonly used as a white pigment in creams, lotions, and powders. With ground mica, it is also used to obtain pearl eye shadows and nail varnishes. $TiO_2$ nanoparticles, as well as in the medical and cosmetics industry, have also been used as an additive in building materials, paints or plaster [37].

Currently, on the market, there are many creams and body lotions in the form of so-called nanoemulsions. They differ from traditional emulsions in the degree of disintegration of the dispersed phase. Nanoemulsions are metastable, i.e., they have low inter-phase voltage, which ensures their high thermodynamic stability. Their advantages include a simple method of production and durability in a wide temperature range. Such products are characterized by high fluidity and low viscosity. It can easily add them to, among others, biologically active substances [37].

The next nanoparticles used in the cosmetics industry are nanocapsules, also known as nanocarriers. These are colloidal, vesicular systems in which the active substance is in a core surrounded by a polymer coating or is absorbed on its surface. Nanocapsules are characterized by the controlled release of the active agent through the slow enzymatic degradation of the shell polymer. Lipophilic substances that can be encapsulated in the liposome in liquid or gel form are more easily encapsulated. The coating protecting the active substance should be made of natural, bioavailable and biodegradable material. The material used must be durable, in order to protect the enclosed substance from harmful external factors [30,38]. The most commonly used polymer is chitosan, as well as cyclodextrin.

There is a growing trend towards the use of nanotechnology in the cosmetics industry, where most of the leading manufacturers around the world use nanotechnology in many of their products. The Nanotechnology Products Database has collected data on 836 cosmetic nanoproducts of various types. These products are introduced to the world markets by 223 companies with headquarters in 29 different countries [39]. The products are classified as skin care, makeup, UV protection, hair care, personal care, sanitizing, and shaving preparations (see Table 1). Among the numerous cosmetic companies developing the nanocosmetics market, L'Oreal, Procter & Gamble, Henkel, Unilever, Koa Corp, Avon, Shiseido, Beiersdorf, Estee Lauder and Johnson & Johnson are the top 10 companies in terms of the number of nanotechnology-related patents [40]. Other giants of the beauty industry are Lancôme, Freeze 24/7, Colorescience, Doctor's Dermatologic Formula, Dermaswiss, Zelens and Euoko. They all delved into the use of nanomaterials to manufacture their products [38]. Testing commissioned by Friends of the Earth Australia has found nanoparticles in foundations and concealers sold by 10 top name brands including Clinique, Clarins, Revlon, The Body Shop, Max Factor, Yves Saint Laurent and Christian Dior [41].

**Table 1.** Nanomaterials in cosmetic companies [41,42].

| Nanomaterials | Type of Cosmetics | Manufacturer |
| --- | --- | --- |
| Zinc oxide, aluminium oxide, iron oxide and titanium dioxide | Mineral Foundation | By Terry<br>Max Factor<br>The Body Shop |
| | Foundation | Christian Dior<br>L'Oreal<br>Clarins |
| | Concealer | Clinique<br>Lancôme Paris<br>Revlon<br>Yves Saint Laurent |
| | UV protection | ColoreScience<br>Dermatone<br>Procter& Gamble<br>Boots |
| Fullerenes and fullersomes | Night and eye cream | Dr. Brandt<br>Sircuit cosmeceuticals<br>Bellapelle skin studio |
| Nanoemulsions | moisture mist<br>Calming nanoemulsion | Chanel<br>La prairie |
| Nanocapsules | skin cream | Dr. Brandt<br>Lancome<br>Enprani |
| Novasomes | Linia Neutrogena<br>Renutriv range,<br>resilience range | Johnson& Johnson<br>Estee lauder |
| Nano silicon dioxide | lift makeup | Lancome |

## 2. Environmental and Health Risk Resulting from the Use of Nanoparticles

Along with the increase in the production and use of nanoparticles in our daily lives, their spread in water systems is observed through the discharge of industrial sewage, municipal sewage treatment plants or surface runoff to the soil, among others. Their release into the environment occurs during production, transport, consumption, and disposal (Figure 2).

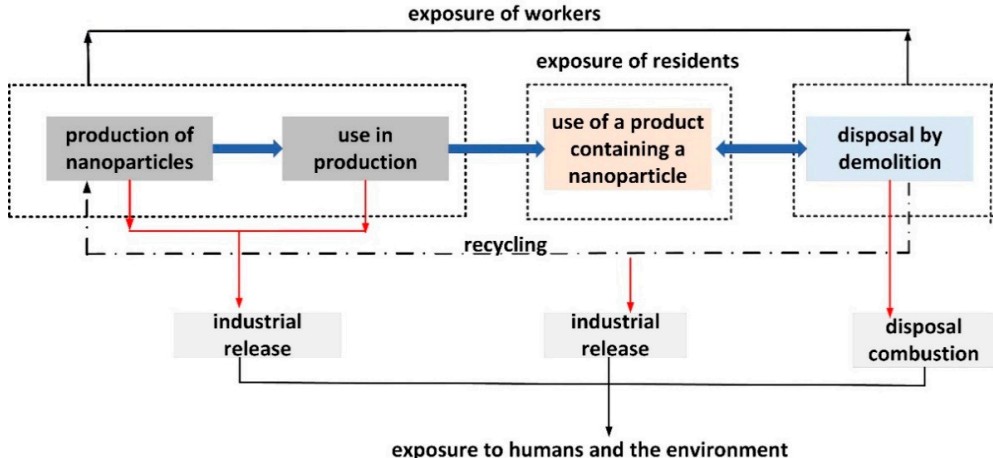

**Figure 2.** Release of nanoparticles to the environment, modified from [43].

Pollution of individual components of the environment with nanoparticles can have point character (i.e., plants producing nanoparticles or nanoproducts, waste incineration plants, waste landfills or sewage treatment plants) or surface (area). Most surface contaminants are associated with the release of nanoparticles during use. In addition to the unintentional release of NP into the environment, we also have the problem of deliberate introduction. An example is NPs being injected directly into groundwater contaminated with nZVI chlorinated solvents. In the environment, the formation of aggregates, and therefore of larger particles that are trapped or eliminated through sedimentation, affects the concentrations of free nanoparticles (Figure 3) [6]. Nanoparticles may pose an ecotoxicological risk in natural receivers and their bioaccumulation in the natural environment and potential inclusion in food chains may also affect human health [4,30,36]. People can either be directly influenced by NPs by exposure to air, soil or water, or indirectly by consuming accumulated plants or animal NPs. Aggregated or adsorbed NPs will be less mobile, but collection by sedimented animals creates a risk of inclusion in the food chain [6].

Silver nanoparticles are released relatively easily (e.g., during the washing of the clothing they contain) and $TiO_2$ (e.g., as a result of their being washed from the building facade from the paint they contain). In the environment, nanoparticles can undergo many different transformations, the nature of which is influenced by both the properties of nanomaterials and the type of receiving medium. From this point of view, nano waste management is a new challenge. This issue emphasizes the need for continuous monitoring of the fate of nanoproducts and suggests the use of recycling as a way to reduce their quantity [34–37]. In addition, research into the disposal of nano waste is necessary to limit the unintentional release of nanomaterials into the environment [37]. Nanomaterials released into the environment can react with the components of air, water, and soil, which causes changes in particle load, surface properties or the ability to form aggregates, among others [38]. The American National Research Council pointed out that research in nanomaterials should focus on identifying the so-called "critical interaction elements" that are necessary to assess the exposure, hazards and thus the risk posed by the designed nanomaterials. These critical elements include chemical and biological-physical transformations that ultimately affect the stability of nanomaterials, their bioavailability/absorption and their reactivity [43].

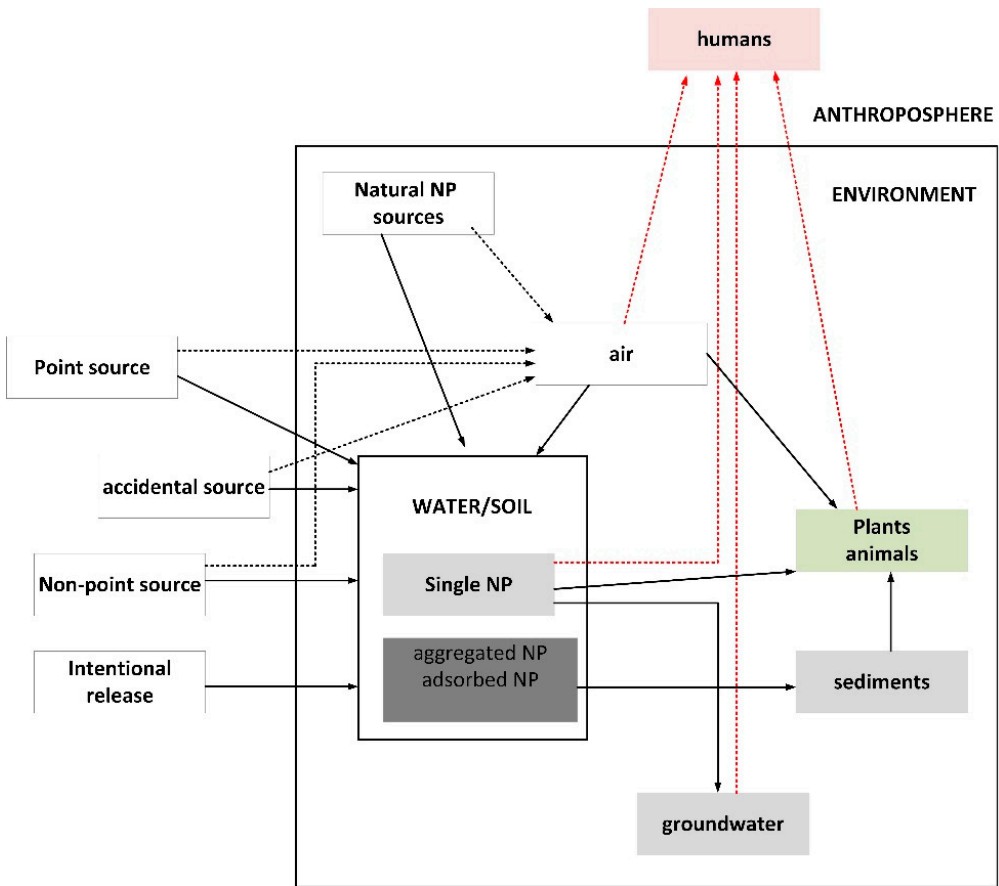

**Figure 3.** Nanoparticle pathways from the anthroposphere into the environment, modified from [6].

According to the Globally Harmonized System, aquatic toxicity can be expressed in five classes:

- extremely toxic < 0.1 mg/L,
- very toxic 0.1–1 mg/L,
- toxic 1–10 mg/L,
- harmful 10–100 mg/L,
- non-toxic > 100 mg/L.

When determining the adverse impact of NPs on health, there is talk of the "3Ds" (i.e., dimension, dose, and durability) [44]. The correlation of NPs' toxicity and their physicochemical properties is shown in Figure 4.

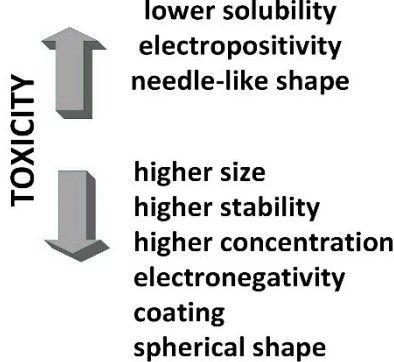

**Figure 4.** The correlation of NPs' toxicity and their physicochemical properties, modified from [44].

After contact with live cells, there are three signs of nanoparticle toxicity: chemical toxicity, small size, and shape. Additionally, the toxicity of nanoparticles is affected by mass, number, size, mass or surface chemistry, aggregation, and stability. The method of administration and the exposure time also influence the severity of NP toxicity [45–48]. So far, the impact of NPs on human health, as well as their impact on the environment, has been relatively little-studied. In the case of nanoparticles, there are four routes of exposure, i.e., respiratory, alimentary, parenteral and dermal (Figure 5). Most nanoparticles enter the body through the respiratory tract, and this applies mainly to those with a size of 10–20 nm. Their impact on health depends on the time spent in the airways and the load on the lungs. Smaller particles have higher toxicity than larger particles with the same composition and crystalline structure. They also generate higher inflammation [45,49,50]. From the lungs, they easily penetrate to other internal organs. Due to their small size, they can: penetrate the blood–brain barrier (BBB), accumulate in the nervous system, cause inflammation and oxidative stress, and cause other various health problems in humans and animals [45,51,52]. Inhalation of asbestos-like fibers and fine dust, though of low toxicity (such as $TiO_2$ NP), may also be associated with chronic inflammatory processes. The NM carcinogenic potential may be different depending on specific properties, such as their reactivity, retention time and distribution in the body, which determine their toxicity in terms of quality [53]. Small carbon nanotubes (< 2.5 μm) and the finest (< 100 nm) $TiO_2$ particles induce tumors in the airways of sensitive animal models when present in high concentrations [38,54–57].

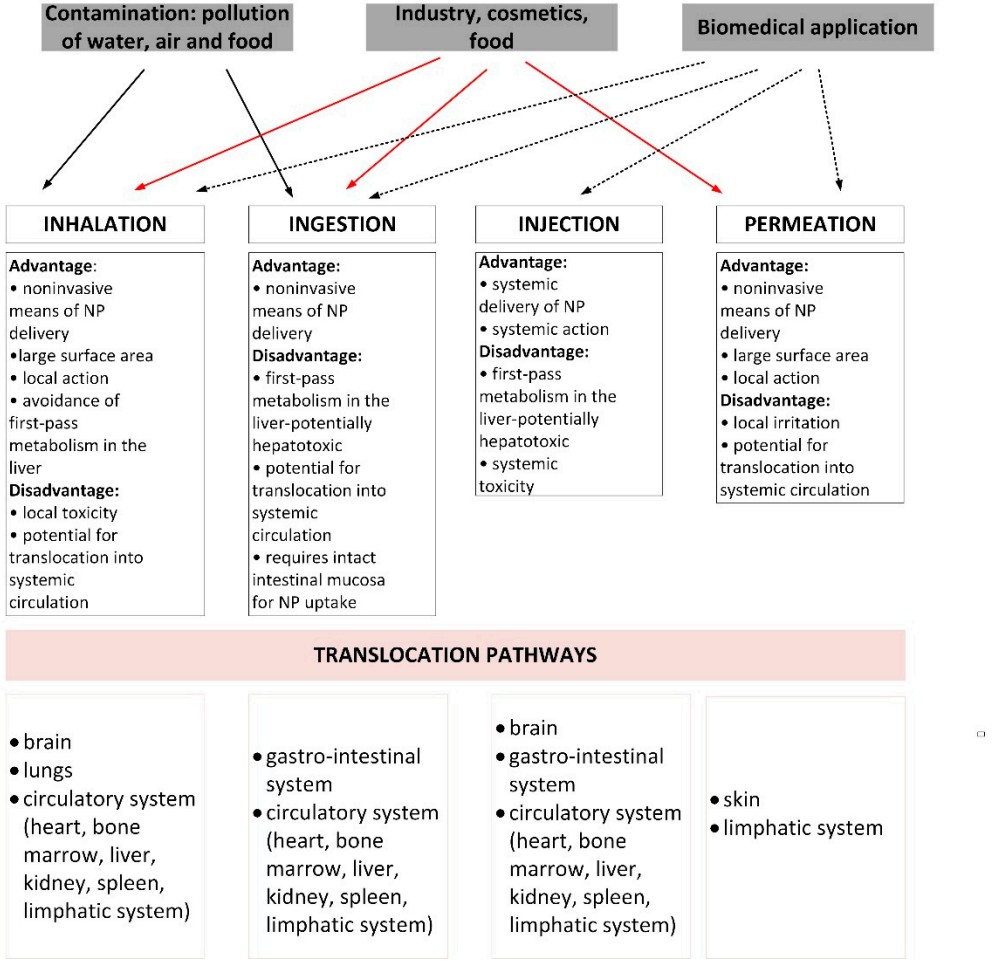

**Figure 5.** Sources of nanoparticles and their ways of entry into the human body, modified from [9,49,54].

Initially widely used in sunscreen, $TiO_2$ was considered to be biologically inert in humans and animals. However, recent studies have shown adverse effects [58–62]. $TiO_2$, which is quite inert as a bulk material, becomes extremely dangerous when long, wire-shaped, and fibrous. Titanium dioxide can produce toxic oxygen radicals (ROS) in the body, which cause oxidative stress in the cells. Numerous studies have shown that $TiO_2$ is genotoxic and carcinogenic. It damages the structure of cells and penetrates the cell nucleus. The widespread use of this compound may be one of the causes of the epidemics of cancer currently observed around the world, as well as degenerative diseases and reproductive disorders. This compound also acts as neurotoxin, damaging nerve cells and leading to their neurodegeneration. Zinc oxides and silver nanoparticles cause the greatest damage to human DNA [62–65]. Cytotoxicity, membrane damage, and increased oxidative stress have been reported in various mammalian cell lines as the most common toxic effect of zinc-based nanomaterials. In contrast, silicon dioxide, usually added during the production of food and medicine, and iron oxide and cerium oxide have low toxicological properties. Further research by scientists will also check to what extent metal oxide nanoparticles can be dangerous to humans [66–68].

According to the reports of the Scientific Committee for Emerging and Newly Identified Health Risks (SCENIHR), and the European Agency for Safety and Health at Work [69], not all nanomaterials are toxic. In order to understand well how nanomaterials work in a new product placed on the market, a case-by-case approach should be taken. As a result of the conducted research, the most significant influence of nanomaterials was found to be in the lungs, including inflammation and tissue damage, pulmonary fibrosis and cancer. Nanomaterials can also have an effect on the circulatory system. Some types of carbon nanotubes can even have similar effects to asbestos. As with the lungs, nanomaterials can attack other organs and tissues, including the liver, kidneys, heart, brain, skeleton, and soft tissues.

## 3. Legal and Environmental Regulations in the Field of Nanotechnology

The difficulties in detecting nanomaterials in cosmetics, food, waste, soil, water, etc. are due to the low concentration of these materials and, above all, to the lack of comparable data and toxicity limits for individual types of nanoparticles. Additional problems regarding the possibility of assessing the degree of exposure of the environment and living organisms to nanoparticles result from insufficient data that could have been provided as a result of systematic detailed tests, a lack of information on possible exposure and existing dangers resulting from prolonged contact, a lack of standards helpful in risk assessment and a lack of adequate protection. Most of the work being carried out in the safe use of nanotechnology is at the planning stage, adapting certificates and standards. As part of the recommendations from the European Commission's science and knowledge service in the last decade, standards for research methods of nanomaterials include [70]:

- ISO/TS 10798:2011-Nanotechnologies—Characterization of single-wall carbon nanotubes using scanning electron microscopy and energy dispersive X-ray spectrometry analysis
- ISO/TS 10797:2012-Nanotechnologies—Characterization of single-wall carbon nanotubes using transmission electron microscopy
- ISO/TS 10868:2017-Nanotechnologies—Characterization of single-wall carbon nanotubes using ultraviolet-visible-near infrared (UV-Vis-NIR) absorption spectroscopy
- ISO/TR 11251:2019-Nanotechnologies—Characterization of volatile components in single-wall carbon nanotube samples using evolved gas analysis/gas chromatograph-mass spectrometry
- ISO/TS 11308:2020-Nanotechnologies—Characterization of carbon nanotube samples using thermogravimetric analysis
- ISO/TS 13278:2017-Nanotechnologies—Determination of elemental impurities in samples of carbon nanotubes using inductively coupled plasma mass spectrometry
- ISO/TS 18827:2017-Nanotechnologies—Electron spin resonance (ESR) as a method for measuring reactive oxygen species (ROS) generated by metal oxide nanomaterials

- ISO/TS 19590:2017-Nanotechnologies—Size distribution and concentration of inorganic nanoparticles in aqueous media via single particle inductively coupled plasma mass spectrometry
- ISO/TS 19807-1:2019-Nanotechnologies—Magnetic nanomaterials—Part 1: Specification of characteristics and measurements for magnetic nanosuspensions
- ISO/TS 21356-1:2021-Nanotechnologies—Structural characterization of graphene—Part 1: Graphene from powders and dispersions.

According to the data in [71]: for the analysis of nanomaterials, transmission and scanning microscopy and atomic force microscopy are used; for the analysis of the size of nanoparticles and agglomerates, fluorescence and plasma spectroscopy, Raman spectrometry are used; absorption is used for single-walled carbon nanotubes analysis; for composition analysis, chemical materials, gas and laser porosimeters are used; for the analysis of pore distribution, the size of the active surface, particle size, elemental analyzers (X-ray Diffraction analyzer) and chromatographs to determine the chemical composition of nanomaterials. Additional analyses of nanomaterials are performed using electrochemical methods, based on the observation of electrocatalytic properties of nanoparticles, viscosimeters. It is also necessary to develop research methods and adapt the measuring equipment that allows the analysis of the mass, number, surface area and particle size distribution of nanomaterials. Collecting this data will lead to obtaining a full picture of the operation of a given type of nanomaterial.

The most important barriers in managing the risk of nanomaterials are the difficulties in developing technologies for detecting these materials—including portable devices for rapid diagnosis —and measuring their concentration and characterizing the degree of surface development, chemical composition, and origin. According to the resolution of the European Parliament "Regulatory aspects of nanomaterials", the use of nanomaterials should be guaranteed to the public, while guaranteeing safety [71]. Safety resulting from materials technology at the nanoscale is determined by legal mechanisms, monitoring, and, above all, knowledge about the thresholds of the harmfulness of nanomaterials to the environment [56]. There is still no information about the tests, and those that are used today are based on the number of nanoparticles. Meanwhile, the toxicity of these materials increases with decreasing dimensions.

In 2012, in a communication to the European Parliament, the European Commission decided to introduce a uniform definition of nanomaterials in the EU and to introduce it into the legislation of the Member States [69–71]. An important legal regulation in the use of nanomaterials is split into two regulations: REACH (*Registration, Evaluation, and Authorization of Chemicals*) and CLP (*Classification, Labelling and Packaging*) [72,73]. When analyzing the content of the ordinances, there is no order to carry out specific tests for individual characters. Decision 2011/381/EU [74] indicates that no product may contain ingredients that meet the criteria:

- in the hazard class "carcinogenicity" category 1A or 1B in accordance with the CLP Regulation;
- in the hazard class "germ cell mutagenicity" category 1A or 1B in accordance with the CLP Regulation;
- in the hazard class "reproductive toxicity", including reproductive function and fertility, or on development of categories 1A or 1B in accordance with the CLP Regulation;
- persistent and very persistent substances, bioaccumulative and toxic according to the criteria described in REACH;
- other endocrine disruptors, the substances listed above and others for which there is scientific evidence of likely serious effects on human health or the environment.

In 2012, 10 EU countries prepared a letter to the European Commission regarding the creation of new legislation [75]. In the absence of appropriate action, some countries have started work on their own. Table 2 shows the countries that have taken the individual initiative in the field of legislative actions on the use of nanomaterials. The most important achievements include the work of the RIVM group in the Netherlands, which has developed nano reference values (NRV).

**Table 2.** Legislative actions in individual countries regarding the use of nanomaterials [75].

| Country | Type of Activities | New Activities |
|---|---|---|
| Denmark's | Environmental Protection Agency decided that nanomaterials should be registered | Guideline for the Danish Inventory of Nanoproducts-2014 |
| France | In 2013, it introduced a decree on the content and conditions for submitting annual declarations covering substances in the form of nanoscale | Not fund |
| Belgium | A project is being developed, based on which reporting will be introduced in line with the quantitative limits of nanomaterials | Royal Decree amending the Royal Decree of May 27th 2014 concerning the placing on the market of substances produced in nanoparticle state-2017 |
| Canada | A review of the chemical law is underway to adapt it to the use of nanomaterials, the first standard for workplace nanotechnology has been developed based on ISO/TR 12885 [70] | New Substances Program Advisory Note-2014 |
| Netherlands | Introduced proposals for risk assessment and setting acceptable levels as part of the work of the National Institute of Public Health and the Environment (Rijksinstituut voor Volksgezondheid en Milieu RIVM) | The European Union Observatory for Nanomaterials–National Institute for Public Health and the Environment-2017 |
| USA | National Institute for Occupational Safety and Health—NIOSH | National Nanotechnology Coordination Office (NNCO) |
| Japan | The project of the Organization for the Development of New Energy and Industrial Technologies in Japan (NEDO) concerns the risk assessment of manufactured nano-objects: titanium dioxide, fullerene and carbon nanotubes | Not found |

Recently, the concerned countries have continued their activities in the field of control and evaluation of nanomaterials performance in individual areas of life and the environment. The Danish Parliament has decided to establish an inventory of mixtures and products that contain or release nanomaterials. The legal framework for this inventory is described in Statutory Order No. 5, 2014 [76] (Table 2). At the request of the interdepartmental working group on risks related to nanomaterials, the Dutch government (IWR) has been commissioned to conduct a study on the risks of nanotechnology, as published in the European Union Observatory for Nanomaterials in 2017 [77]. In the US, the National Nanotechnology Coordination Office (NNCO) is the primary point of contact for information on NNI; it provides technical and administrative support and promotes access to and an early application of technology, innovation and expertise derived from NNI activities [78]. In 2014, Advisory Note 2014-02 for the New Substances Scheme was published in the Canadian evaluation of nanomaterials in new substances "Rules for notification" (chemicals and polymers). The purpose of this advisory note is to inform Canadian manufacturers and importers that new substances in the nano size range (1–100 nanometers) must be notified under the new substance notification provisions (chemicals and polymers) [79].

The reference values for nanomaterials are expressed as values corresponding to the weighted average operating hours of 8 h and as instantaneous values corresponding to the weighted average operating hours of 15 min. These values set the warning level; when reference values are exceeded, appropriate exposure controls should be used [80–82]. Table 3 shows the breakdown of nanomaterials into individual hazard classes by NRV.

**Table 3.** The degree of toxicity of nanomaterials and hazard classes depending on the type of material [83–86].

| Toxicity | NRV Hazard Class | Reference Values | Type of Nanomaterial | Example |
|---|---|---|---|---|
| high | class 1 | 0.01 fibers cm$^{-3}$ | rigid carbon nanofibers, metal oxide fibers | SWCNT (single-walled carbon nanotubes) or MWCNT (multiwalled carbon nanotubes), fullerenes |
| medium or low | class 2a | 20,000 particles cm$^{-3}$ | granular nanomaterials (non-fibrous), stable in the environment, with a density greater than 6 g cm$^{-3}$ | particles Ag, Au, CeO$_2$, COO, Fe, FexOy, La, Pb, Sb$_2$O$_5$, or SnO$_2$ |

**Table 3.** *Cont.*

| Toxicity | NRV Hazard Class | Reference Values | Type of Nanomaterial | Example |
|---|---|---|---|---|
| medium or low | class 2b | 40,000 particles $cm^{-3}$ | granular nanomaterials and nanofibers, stable in the environment, with a density above 6 g $cm^{-3}$ | particles $Al_2O_3$, $SiO_2$, $TiO_2$, ZnO, $CaCO_3$, layered aluminosilicate, carbon black, C60, dendrimers, polystyrene or nanofibers |
| low | class 3 | OEL values * | granular nanomaterials, unstable or soluble in water (solubility above 100 mg $L^{-1}$) | NaCl, lipid particles, flour, sucrose. |

\* OEL—occupational exposure levels.

## 4. Problems and Challenges in the Application of Nanotechnology in Environmental Engineering

Products in the form of drugs and cosmetics easily enter into the environment. In order to protect the environment against the ingress of nanoscale active materials, the level of nanomaterials in products, especially those newly introduced to the market, should be controlled. The USA plays an important role in the safety assessment and monitoring of the performance of cosmetics and drugs through continuous product reviews by the Food and Drug Administration (FDA) [80]. For this purpose, the agency has developed a special research program aimed at obtaining as many tools and methods as possible to identify the properties of nanomaterials and their impact on products. The agency takes a conservative scientific approach to assessing each product for its merits and does not make broad, general assumptions about the safety of nanotechnology products. In 2006, the FDA's nanotechnology working group assessed and identified possible knowledge or policy gaps to enable the agency to better assess the safety aspects of FDA-regulated products. In 2007, the Nanotechnology Working Group published a report, which indicated recommendations for actions that the agency may take in implementing its mission of protecting and promoting public health [81]. With the increasing number of submissions of products containing nanomaterials, the task force strongly encourages internal research grants, provides rapid training in nanotechnology, and encourages active participation in the development of international nanotechnology standards. FDA supports the National Nanotechnology Initiative (NNI) and collaborates with other agencies through participation in the Nanoscale Science Engineering and Technology (NSET) subcommittee and the Nanotechnology Environmental and Health Implications (NEHI) working group.

Since the publication of this report, the FDA has issued several guidance documents on topics related to the use of nanotechnology in FDA-regulated products [80]. The guidelines contained in it do not create or confer any rights; they represent the current views of the FDA. In its 2014 reports, the agency does not categorically evaluate nanotechnology, so it does not indicate that nanotechnology is inherently safe or harmful. Guidance takes into account the specific characteristics and effects of nanomaterials in the specific biological context of each product and its intended use. Specific approaches for each product area with nanomaterials (cosmetics [82], drugs, food products [83]) differ in scope and issues; for example, interactions of nanomaterials with natural systems, research approaches, product safety assessments, product quality, etc. from the Guidance for Industry Safety of Nanomaterials in Cosmetic Products [84], and safety is determined by an extensive assessment of physical and chemical properties and by assessing contaminants, if any. The scope and impact of the possible toxicity of nanomaterials is described by indicating the ways of exposure, absorption and penetration into organisms. Additionally, as indicated in the Guidance [84], toxicological studies in vitro and in vivo, clinical studies, and toxicokinetics and toxicodynamics should be considered. Each cosmetic product should therefore contain a whole package of data and information justifying the safety of the product in terms of its conditions of use.

Medicines are subject to stringent FDA-imposed controls for approval, but there are no such requirements for cosmetics. Cosmeceuticals are products on the border of cosmetics and pharmaceuticals. As of today, there is no rigorous control over the approval and

regulation of nanocosmetics. No clinical trials are required for their approval, raising concerns about toxicity after use. Many cosmeceuticals change the physiological processes in the skin, but manufacturers still avoid clinical trials or making specific claims to avoid subjecting their products to costly and lengthy FDA approval processes. If the FDA determines that there is a problem with the safety of any cosmetic or ingredient, the FDA has the power to prohibit the sale and manufacture of the product [85]. In the European Union, cosmetics are subject to the provisions of the Cosmetics Directive 76/768/EEC. The EU doesn't have a named category for cosmeceuticals, but it does have strict rules where any company claims must be presented as evidence. According to the new EU regulation, manufacturers have to replace the nanoparticles contained in the product. Cosmetic regulations state that any product containing nanomaterials as an ingredient should be clearly listed and must insert the word "nano" in the brackets after the list of ingredients [86]. The gap between basic research concerns the impact of nanomaterials that enter into the environment on plants, animals and, consequently, on human health. The phytotoxicity of nanomaterials is currently a topic that is not fully understood and researched. Most of the research to date has focused on germination, cell culture and genetic effects [87]. There are studies showing an increase in the level of reactive oxygen species (ROS) in cells of higher plants under the influence of contact with nanoparticles, which, depending on the dose, resulted in cell death.

Monitoring the presence of nanoparticles is associated with facing the challenges arising from the nature of nanomaterials about the phenomena they undergo or the lack of specialized technique [88,89]:

- chemical properties: high chemical reactivity, increased corrosion resistance, diversity of chemical and phase composition;
- physical properties: small size, and at the same time a high tendency towards aggregation and/or agglomeration, diffusivity, large surface area compared to volume, which results in the appearance of strong sorption properties (adsorption and absorption) and an increase in the catalytic activity of nanomaterials;
- mechanical properties: hardness, abrasion resistance, superelasticity phenomenon occurring as a result of reducing the grain size of intermetallic cluster connections to the order of nanometers;
- biological properties: strong antibacterial properties, penetration through biological barriers, large range of impact due to dimensions and diffusivity.

Assessing the toxicity and safety of nanoparticles requires an understanding of their uptake by organism. Most studies focus on determining the nature of the phytotoxicity of nanoparticles, but quantitative methods for measuring them in plant or animal tissues have not been established. Nanoparticles larger than the pore size of the cell wall stick to the cells of the root epithelium, causing physical damage, clogging the pores and reducing the absorption of water and nutrients [90]. Through the pores in the cell walls, nanoparticles can easily penetrate. It is about their accumulation in plant tissues. As plants are an important food source for humans, further research is needed to evaluate the toxicity caused by nanomaterials. The mechanisms of metal-induced carcinogenesis are not well understood. Both genetic and non-genetic factors induced by nanoparticles in cells may predispose to carcinogenicity [91]. It is imperative to conduct research into the toxicity and genotoxicity of nanoparticles, in order to be able to safely take advantage of the enormous potential benefits of this new technology [92].

## 5. Summary and Future Perspectives

Countless uses of materials at the nanoscale, some of which are presented in this paper, indicate the attractiveness of these materials. $TiO_2$ nanoparticles, as well as the medical and cosmetics industries have also been used as an additive to building materials, paints, and plaster. Titanium oxide nanoparticles and zinc oxide constitute a physical solar filter, impervious to UVA and UVB rays. Silver nanoparticles, due to their inactivating properties of pathogens, are included in fabrics, dressings, food storage containers, and children's

toys. The FDA's New Drug Application (NDA), Investigation New Drug (IND), Center for Devices and Radiological Health (CDRH), and Center for Veterinary Medicine (CVM) control and assess the safety of nanomaterials based on the characteristics of the nanomaterial itself and toxicological analyzes [82,84]. The FDA takes into account very broadly the impact of nanomaterials and the areas of their use. In the case of cosmetics, and medical products that are going to be placed on the market and have new or changed properties, the need for safety testing needs to be assessed. For food products, the assessment is made on a case-by-case basis. The FDA recommends a safety assessment taking into account, e.g., physicochemical properties of nanomaterials, agglomerations and size distribution of nanoparticles, the presence and impact of possible pollutants, potential routes of exposure to nanomaterials. In terms of toxicity, it is recommended to conduct tests to obtain in vitro and in vivo toxicological data of nanomaterial components and their impurities, skin penetration, potential inhalation, irritation (skin and eyes) and sensitization tests, mutagenicity/genotoxicity tests.

The development of production, application areas and exploitation of materials at the nanoscale constantly increase the risk of the potential releasing of these materials into the environment [93,94]. According to one theory regarding the mechanism of action of nanoparticles on microbes, it follows that nanoparticles, due to their size, have a high penetration capacity, thus causing irreversible structural changes in living organisms, and consequently a negative impact on the environment and human health. The most important barriers in managing the risk of nanomaterials are the difficulties in developing technologies for detecting these materials, including portable devices for rapid diagnosis, measuring their concentration and characterizing the degree of surface development, chemical composition, and origin. According to the resolution of the European Parliament "Regulatory aspects of nanomaterials", it is important to ensure that the public uses nanomaterials while guaranteeing safety. Safety arising from material technology at the nanoscale is determined by legal mechanisms, monitoring and, above all, knowledge about the thresholds of nanomaterial harmfulness to the environment. There is still no information about the tests, and those that are used today are based on the number of nanoparticles. Meanwhile, the toxicity of these materials increases with decreasing dimensions. The spread of nanoparticles in water systems occurs, among other ways, through the discharge of industrial wastewater, to municipal sewage treatment plants or through surface runoff to soils, and their release into the environment occurs during production, transport, consumption, and disposal. The problem became so important that, in 2012, in a communication to the European Parliament, the European Commission decided to introduce a uniform definition of nanomaterials in the EU and to introduce it into the legislation of the Member States, and two REACH and CLP regulations supplement this important legal regulation. It is noteworthy that only a few countries have taken legislative action regarding the use, monitoring and storage conditions of nanomaterials.

Nanotechnology-based products pose significant challenges for governments, relevant ministries and industry to ensure consumer confidence and acceptance. Nanoscale materials are produced worldwide, but very few countries have standard regulatory rules for the industrial use of nanotechnology. Especially the lack of control over the behavior of nanomaterials in the environment and the monitoring of their transmission in the natural environment. Insufficient research on nanosystems is making it difficult to draw conclusions about the effects of nanotechnology development in the world. The use of nanoparticles poses a hazard and risk that nanomaterials may enter the food chain through air, water and soil during their manufacture and use, leading to DNA damage, cell membrane disruption and cell death. It should be part of global policy to ensure appropriate labeling and rules are recommended when placing nanoproducts on the market, which can help increase consumer acceptance. The use of nanotechnology, with proper management and regulation, can play a significant role in improving the quality of life, benefiting people's health and well-being.

**Author Contributions:** Idea of manuscript, R.W.; writing—original draft preparation, R.W., A.K.-K.; writing—review and editing, supervision, R.W. and A.K.-K. All authors have read and agreed to the published version of the manuscript.

**Funding:** The scientific research was funded by the statute subvention of Czestochowa University of Technology, Faculty of Infrastructure and Environment. The research was funded by the project No. BS/PB400/301/21.

**Institutional Review Board Statement:** Not applicable.

**Informed Consent Statement:** Not applicable.

**Data Availability Statement:** Not applicable.

**Conflicts of Interest:** The authors declare no conflict of interest.

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
