# Peer review of "Nanoparticles from the Cosmetics and Medical Industries in Legal and Environmental Aspects"

_sustainability, doi:10.3390/su13115805_

Round 1
Reviewer 1 Report
Manuscript No sustainability-1197211 is a literature review paper on nanoparticles, their uses, toxicity, and legislation. This study is well organized and presented. Some suggestions for authors are given below :
- The text needs a revision from a native speaker.
- The addition of more references dated from the last 5 years should be added
Author Response
Answer to Reviewer #1
Thank you very much for all your comments and time. Below we present the changes that were made to the manuscript and the responses to the remarks of the honorable Reviewer:
Manuscript No sustainability-1197211 is a literature review paper on nanoparticles, their uses, toxicity, and legislation. This study is well organized and presented. Some suggestions for authors are given below :
- The text needs a revision from a native speaker.
Manuscript text has been edited by a native speaker:
- Line 52: NPs are characterized by a specific geometric structure, characterized by a high surface-to-volume ratio, which is greater the smaller the diameter of the particles
- Change to: NPs are characterized by a specific geometric structure, distinguished by a high surface-to-volume ratio, which is greater the smaller the diameter of the particles
- Line 60: NPs generally have significantly different physicochemical properties compared to small particles of the same composition.
- Change to: In general, NPs have significantly different physicochemical properties compared to small particles of the same composition
- Line 66: Their number is estimated at over 1000. Change to: Their number is evaluated at over 1000.
- Line 71: The mentioned directions of the application of nanoparticles are presented in Fig. 1. Change to: The mentioned directions of nanoparticles application are presented in Fig. 1.
- Line 79: The antibacterial properties of silver nanoparticles result from their small size and developed surface, which allows them to easily penetrate biological membranes and penetrate microorganisms, causing them to die [24]. Change to: The antibacterial properties of silver nanoparticles result from their small size and developed surface, which allows them easily penetrating biological membranes and microorganisms, causing their death [24].
- Line 90: The return to the use of silver as a bactericide in the form of solutions, suspensions or nanoparticle forms has its justification because deposited on various supports, e.g. silica or polymeric, are considered one of the most effective disinfectants combining the biocidal and deodorizing properties of silver and silicon. Change to: The return to the use of silver as a bactericide in the form of solutions, suspensions or nanoparticle forms has its justification as it is deposited on various substrats, e.g. silica or polymeric, are considered one of the most effective disinfectants combining the biocidal and deodorizing properties of silver and silicon.
- Line 97: Gold in the form of a nano shows great ease of penetration into the cells and acts strongly regenerating and stimulating [25]. Change to: Gold in the nano form shows great ease of penetration into the cells and acts strongly regenerating and stimulating [25].
- Line 99: In cosmetics, zinc oxide and titanium (IV) oxide are most widely used. Change to: In cosmetics, zinc oxide and titanium (IV) oxide are most commonly used.
- Line 99: These types of compounds are components of so-called mineral cosmetics, which in recent years increasingly take the place of traditional ones. Change to: These types of compounds are components of so-called mineral cosmetics, which have been increasingly replacing traditional in recent years.
- Line 104: The presence of NP is to influence, among others to reduce the effects of photoaging, reduce wrinkles and discoloration, and even prevent graying and reduce hair loss. Change to: The presence of NP is to influence, among others to reduce the effects of photoaging, wrinkles and discoloration, reduce hair loos and even prevent graving.
- Line 112: Zinc oxide is obtained by micronization, i.e. fragmentation to fine pollen, with a grain size of about 20 nm, and in this form, it is suspended in silicone powder. Change to: Zinc oxide is obtained by micronization, i.e. fragmentation to fine pollen, with a grain size of about 20 nm in this form it is suspended in silicone powder.
- Line 123: Paradoxically, therefore, cosmetics containing titanium dioxide instead of inhibiting, accelerate the aging process. Change to: Therefore, paradoxically, cosmetics containing titanium dioxide instead of inhibiting it, accelerate the aging process.
- Line 131: It differs from traditional emulsions in the degree of disintegration of the dispersed phase. Change to: They differ from traditional emulsions in the degree of disintegration of the dispersed phase.
- Line 133: Their advantages include a simple method of their production, durability in a wide temperature range. Change to: Their advantages include a simple method of their production, and durability in a wide temperature range.
- Line 189: In the environment, nanoparticles can undergo many different transformations, whose nature is influenced by both the properties of nanomaterials and the type of receiving medium. Change to: In the environment, nanoparticles can undergo many different transformations, which nature is influenced by both the properties of nanomaterials and the type of receiving medium.
- Line 194: Also, research into the disposal of nano waste is necessary to limit the unintentional release of nanomaterials into the environment [34]. Change to: In addition, research into the disposal of nano waste is necessary to limit the unintentional release of nanomaterials into the environment [34].
- Line 212: none toxic> 100 mg / l. Change to: non toxic> 100 mg / l.
- Line 221: Also, mass, number, size, mass or surface chemistry, aggregation, and stability affect the toxicity of nanoparticles. Change to: Additionally, the toxicity of nanoparticles is affected by mass, number, size, mass or surface chemistry, aggregation, and stability
- Line 222: Also, the method of administration and exposure time influence the severity of NP toxicity [42-45]. Change to: The method of administration and the exposure time also influence the severity of NP toxicity [42-45].
- Line 244: Numerous studies have shown that TiO2 damages the structure of cells, penetrates the cell nucleus, is genotoxic and carcinogenic. Change to: Numerous studies have shown that TiO2 is genotoxic and carcinogenic. It damages the structure of cells and penetrates the cell nucleus.
- Line 248: This compound also acts neurotoxin, damaging nerve cells and leading to their neurodegeneration. Change to: This compound also acts as neurotoxin, damaging nerve cells and leading to their neurodegeneration.
- Line 331: An important legal regulation in the use of nanomaterials is two regulations: REACH (Registration, Evaluation, and Authorization of Chemicals) and CLP (Classification, Lebelling, and Packaging) [72-73]. Change to: An important legal regulation in the use of nanomaterials is split into two regulations: REACH (Registration, Evaluation, and Authorization of Chemicals) and CLP (Classification, Labelling, and Packaging) [72-73].
- Line 446: Monitoring the presence of nanoparticles is associated with taking on the challenges arising from the nature of nanomaterials about the phenomena they undergo or the lack of specialized technique [88-89]. Change to: Monitoring the presence of nanoparticles is associated with facing the challenges arising from the nature of nanomaterials about the phenomena they undergo or the lack of specialized technique [88-89]:
- Line 495: The development of production, application areas and exploitation of materials at the nanoscale constantly increase the risk of the potential release of these materials into the environment [93-94]. Change to: The development of production, application areas and exploitation of materials at the nanoscale constantly increase the risk of the potential releasing these materials into the environment [93-94].
- The addition of more references dated from the last 5 years should be added
The following references have been added from the last 5 years:
- Verma A., Mehata M.S., Controllable synthesis of silver nanoparticles using Neem leaves and their antimicrobial activity, J. Radiation Research and Applied Science, 9, 2016, 109-115.
- Ahmed S., Ikram S., Yudha S.S., Biosynthesis of gold nanoparticles: A green approach, Journal of Photochemistry & Photobiology, B: Biology, 161, 2016, 141-153
- Khan I., Saeed K., Khan I., Nanoparticles: Properties, applications and toxicities, Arabian Journal of Chemistry, 12, 2019, 908-931
- Guidance document for the testing of dissolution and dispersion stability of nanomaterials and the use of the data for further environmental testing and assessment strategies, Series on Testing Assessment No. 318, 9, 2020.
- Chaudhary P., Fatima F., Kumar A., Relevance of Nanomaterials in Food Packaging and its Advanced Future Prospects, Journal of Inorganic and Organometallic Polymer and Materials, 30, 2020, 5180-5192.
- Bilal M., Iqbal H. M. N., New Insights on Unique Features and Role of Nanostructured Materials in Cosmetics, Cosmetics, 7, 2020, 24
- Bilal M., Mehmood S., Iqbal H, The Beast of Beauty: Environmental and Health Concerns of Toxic Components in Cosmetics, 7, 2020, 13
- Effong D.E., Uwah T. O., Jumbo E. U., Akpabio A.E., Nanotechnology in Cosmetics: Basic, Current Trends and Safety Concerns- A Review, Advanced in Nanoparticles, 9, 2020, 1-22.
- Zhao J., Lin M. Wang Z., Cao X., Xing B., Engineered nanomaterials in the environment: Are they safe? Critical Reviews in Environmental Science and Technology, 2020, https://doi.org/10.1080/10643389.2020.1764279
- Mustafa F., Andreescu S., Nanotechnology-based approaches for foof sensing and packaging applications, Royal Society of Chemistry, 10, 2020, 19309, DOI: 10.1039/d0ra01084g
- Nile S. H., Baskar V., Selvaraj D., Nile A., Xiao J., Kai G., Nanotechnologies in Food Science: Applications, Recent Trends, and Future Persectives, Nano-Micro Letters, 12, 2020, 45, https://doi.org/10.1007/s40820-020-0383-9
- Nanotechnology-Over a Decade of Progress and Innovation, A Report by The U.S. Food and Drug Administration, July 2020.

Reviewer 2 Report
Page 3, Line 88 - you mentioned catheters twice
Page 7, Figure 5 - sysytem instead system.
In 4. Problems and challenges... You mentioned the lack of specialized technique. Please check the informations on new devices and tests available to detect these nanomaterials in air, soil, water. Is there any progress on new techniques and devices, or is it still just number of particles?
In Table 1. You gave list of countries that took legislative actions. Since this actions were taken in 2012, and now is 2021, was there any progress in given countries since then or on EU level? Please state that in the article.
Out of 65 scientific references only 2 are from 2020. Please add more recent references.
Please give in Summary future perspective on this matter.
Author Response
Answer to Reviewer #2
Thank you very much for all your comments and time. Below we present the changes that were made to the manuscript and the responses to the remarks of the honorable Reviewer:
- Page 3, Line 88 - you mentioned catheters twice
Change to: Currently, silver nanoparticles can be found in fabrics (dressings, socks, underwear), cosmetics (powders, deodorants) and medicine (catheters, implants).
- Page 7, Figure 5 - sysytem instead system.
Change to:
Figure 5. Sources of nanoparticles and their ways of entry into the human body [9, 46, 51]
- In 4. Problems and challenges... You mentioned the lack of specialized technique. Please check the informations on new devices and tests available to detect these nanomaterials in air, soil, water. Is there any progress on new techniques and devices, or is it still just number of particles?
As part of the recommendation of The European Commission's science and knowledge, in the last decade, standards for research methods of nanomaterials, including [67]:
- ISO/TS 10798:2011- Nanotechnologies — Charaterization of single-wall carbon nanotubes using scanning electron microscopy and energy dispersive X-ray spectrometry analysis
- ISO/TS 10797:2012- Nanotechnologies — Characterization of single-wall carbon nanotubes using transmission electron microscopy
- ISO/TS 10868:2017- Nanotechnologies — Characterization of single-wall carbon nanotubes using ultraviolet-visible-near infrared (UV-Vis-NIR) absorption spectroscopy
- ISO/TR 11251:2019- Nanotechnologies — Characterization of volatile components in single-wall carbon nanotube samples using evolved gas analysis/gas chromatograph-mass spectrometry
- ISO/TS 11308:2020- Nanotechnologies — Characterization of carbon nanotube samples using thermogravimetric analysis
- ISO/TS 13278:2017- Nanotechnologies — Determination of elemental impurities in samples of carbon nanotubes using inductively coupled plasma mass spectrometry
- ISO/TS 18827:2017- Nanotechnologies — Electron spin resonance (ESR) as a method for measuring reactive oxygen species (ROS) generated by metal oxide nanomaterials
- ISO/TS 19590:2017- Nanotechnologies — Size distribution and concentration of inorganic nanoparticles in aqueous media via single particle inductively coupled plasma mass spectrometry
- ISO/TS 19807-1:2019- Nanotechnologies — Magnetic nanomaterials — Part 1: Specification of characteristics and measurements for magnetic nanosuspensions
- ISO/TS 21356-1:2021- Nanotechnologies — Structural characterization of graphene — Part 1: Graphene from powders and dispersions.
According to the data in [84], for the analysis of nanomaterials, transmission and scanning microscopy, atomic force microscopy for the analysis of the size of nanoparticles and agglomerates, fluorescence and plasma spectroscopy, Raman spectrometry, Absorption for Single-Walled Carbon Nanotubes Analysis for composition analysis are used chemical materials, gas and laser porosimeters for the analysis of pore distribution, the size of the active surface, particle size, elemental analyzers (X — ray Diffraction analyzer) and chromatographs to determine the chemical composition of nanomaterials. Additional analyzes of nanomaterials are performed using electrochemical methods, based on the observation of electrocatalytic properties of nanoparticles, viscoimeters.
- In Table 1. You gave list of countries that took legislative actions. Since this actions were taken in 2012, and now is 2021, was there any progress in given countries since then or on EU level? Please state that in the article.
Recently, the countries concerned have continued their activities in the field of control and evaluation of the performance of nanomaterials in individual areas of life and the environment. The Danish Parliament has decided to establish an inventory of mixtures and products that contain or release nanomaterials. The legal framework for this inventory is described in Statutory Order No. 5, 2014 [76] (Table 2). At the request of the Interdepartmental Working Group on Risks Related to Nanomaterials The Dutch government (IWR) has been commissioned to conduct a study on the risks of nanotechnology, as published in The European Union Observatory for Nanomaterials in 2017 [77]. In the US, the National Nanotechnology Coordination Office (NNCO) is the primary point of contact for information on NNI; provides technical and administrative support and promotes access to and early application of technology, innovation and expertise derived from NNI activities [78]. In 2014, Advisory Note 2014-02 for the New Substances Scheme was published in Canada Evaluation of nanomaterials in new substances Rules for notification (chemicals and polymers). The purpose of this advisory note is to inform Canadian manufacturers and importers that new substances in the nano size range (1-100 nanometers) must be notified under the new substance notification provisions (chemicals and polymers) [79].
Table 2. Legislative actions in individual countries regarding the use of nanomaterials [75]
|
Country |
Type of activities |
New activities |
|
Denmark's |
Environmental Protection Agency decided that nanomaterials should be registered |
Guideline for the Danish Inventory of Nanoproducts - 2014 |
|
France |
In 2013, it introduced a decree on the content and conditions for submitting annual declarations covering substances in the form of nanoscale |
Not fund |
|
Belgium |
A project is being developed, based on which reporting will be introduced in line with the quantitative limits of nanomaterials |
Royal Decree amending the Royal Decree of May 27th 2014 concerning the placing on the market of substances produced in nanoparticular state- 2017 |
|
Canada |
A review of the chemical law is underway to adapt it to the use of nanomaterials, the first standard for workplace nanotechnology has been developed based on ISO / TR 12885 [67] |
New Substances Program Advisory Note - 2014 |
|
Netherlands |
Introduced proposals for risk assessment and setting acceptable levels as part of the work of the National Institute of Public Health and the Environment (Rijksinstituut voor Volksgezondheid en Milieu RIVM) |
The European Union Observatory for Nanomaterials – National Institute for Public Health and the Environment - 2017 |
|
USA |
National Institute for Occupational Safety and Health – NIOSH |
National Nanotechnology Coordination Office (NNCO) |
|
Japan |
The project of the Organization for the Development of New Energy and Industrial Technologies in Japan (NEDO) concerns the risk assessment of manufactured nano-objects: titanium dioxide, fullerene and carbon nanotubes |
Not found |
- Out of 65 scientific references only 2 are from 2020. Please add more recent references.
The following references have been added:
- Verma A., Mehata M.S., Controllable synthesis of silver nanoparticles using Neem leaves and their antimicrobial activity, J. Radiation Research and Applied Science, 9, 2016, 109-115.
- Ahmed S., Ikram S., Yudha S.S., Biosynthesis of gold nanoparticles: A green approach, Journal of Photochemistry & Photobiology, B: Biology, 161, 2016, 141-153
- Khan I., Saeed K., Khan I., Nanoparticles: Properties, applications and toxicities, Arabian Journal of Chemistry, 12, 2019, 908-931
- Guidance document for the testing of dissolution and dispersion stability of nanomaterials and the use of the data for further environmental testing and assessment strategies, Series on Testing Assessment No. 318, 9, 2020.
- Chaudhary P., Fatima F., Kumar A., Relevance of Nanomaterials in Food Packaging and its Advanced Future Prospects, Journal of Inorganic and Organometallic Polymer and Materials, 30, 2020, 5180-5192.
- Bilal M., Iqbal H. M. N., New Insights on Unique Features and Role of Nanostructured Materials in Cosmetics, Cosmetics, 7, 2020, 24
- Bilal M., Mehmood S., Iqbal H, The Beast of Beauty: Environmental and Health Concerns of Toxic Components in Cosmetics, 7, 2020, 13
- Effong D.E., Uwah T. O., Jumbo E. U., Akpabio A.E., Nanotechnology in Cosmetics: Basic, Current Trends and Safety Concerns- A Review, Advanced in Nanoparticles, 9, 2020, 1-22.
- Zhao J., Lin M. Wang Z., Cao X., Xing B., Engineered nanomaterials in the environment: Are they safe? Critical Reviews in Environmental Science and Technology, 2020, https://doi.org/10.1080/10643389.2020.1764279
- Mustafa F., Andreescu S., Nanotechnology-based approaches for foof sensing and packaging applications, Royal Society of Chemistry, 10, 2020, 19309, DOI: 10.1039/d0ra01084g
- Nile S. H., Baskar V., Selvaraj D., Nile A., Xiao J., Kai G., Nanotechnologies in Food Science: Applications, Recent Trends, and Future Persectives, Nano-Micro Letters, 12, 2020, 45, https://doi.org/10.1007/s40820-020-0383-9
- Nanotechnology-Over a Decade of Progress and Innovation, A Report by The U.S. Food and Drug Administration, July 2020.
- Please give in Summary future perspective on this matter.
Nanotechnology-based products pose significant challenges for government, relevant ministries and industry to ensure consumer confidence and acceptance. Nano-scale materials are produced worldwide, but very few countries have standard regulatory rules for the industrial use of nanotechnology. Especially the lack of control over the behavior of nanomaterials in the environment and the monitoring of their transmission in the natural environment. Insufficient research on nanosystems is making it difficult to draw conclusions about the effects of nanotechnology development in the world. The use of nanoparticles poses a hazard and risk that nanomaterials may enter the food chain through air, water and soil during their manufacture and use, leading to DNA damage, cell membrane disruption and cell death. As part of global policy, it should ensure appropriate labeling and rules recommended when placing nanoproducts on the market, which can help increase consumer acceptance. The use of nanotechnology, with proper management and regulation, can play a significant role in improving the quality of life, benefiting people's health and well-being.

Reviewer 3 Report
- Authors should summarize in Tables of manufacturers employing nanotechnology in their marketed products of both cosmetics and medicines
- Authors should explain why are nanomaterials make a special issue both cosmetic and medicine in safety and health consideration
- Authors should explanation required about perspectives on FDA’s regulation of nanotechnology in cosmetics and medicines
- Authors should public perception of nanotechnology and confidence in FDA
- Authors should brief about existing gap between basic research relating nanomaterials and their application in cosmetics and medicine
Author Response
Answer to Reviewer #3
Thank you very much for all your comments and time. Below we present the changes that were made to the manuscript and the responses to the remarks of the honorable Reviewer:
- Authors should summarize in Tables of manufacturers employing nanotechnology in their marketed products of both cosmetics and medicines
There is a growing trend towards the use of nanotechnology in the cosmetics industry, where most of the leading manufacturers around the world use nanotechnology in many of their products. The Nanotechnology Products Database has collected data on 836 cosmetic nanoproducts of various types. These products are introduced to the world markets by 223 companies with headquarters in 29 different countries [36]. The products are classified as skin care, makeup, UV protection, hair care, personal care, sanitizing, and shaving preparations. Among the numerous cosmetic companies developing the nanocosmetics market, L'Oreal, Procter & Gamble, Henkel, Unilever, Koa Corp, Avon, Shiseido, Beiersdorf, Estee Lauder and Johnson & Johnson are the top 10 companies in terms of the number of nanotechnology-related patents [37]. Other giants of the beauty industry are Lancôme, Freeze 24/7, Colorescience, Doctor's Dermatologic Formula, Dermaswiss, Zelens and Euoko. They all delved into the use of nanomaterials to manufacture their products [38]. Testing commissioned by Friends of the Earth Australia has found nanoparticles in foundations and concealers sold by 10 top name brands including Clinique, Clarins, Revlon, The Body Shop, Max Factor, Yves Saint Laurent and Christian Dior [39].
Table 1. Legislative actions in individual countries regarding the use of nanomaterials [38-39]
|
Nanomaterials |
Type of cosmetics |
Manufacturer
|
|
Zinc oxide, aluminium oxide, iron oxide and titanium dioxide |
Mineral Foundation |
By Terry Max Factor The Body Shop |
|
Foundation |
Christian Dior L’Oreal Clarins |
|
|
Concealer |
Clinique Lancôme Paris Revlon Yves Saint Laurent |
|
|
|
UV protection |
ColoreScience Dermatone Procter& Gamble Boots |
|
Fullerenes and fullersomes |
night and eye cream |
Dr. Brandt Sircuit cosmeceuticals Bellapelle skin studio |
|
Nanoemulsions |
moisture mist Calming nanoemulsion |
Chanel La prairie |
|
Nanocapsules |
skin cream |
Dr. Brandt Lancome Enprani |
|
Novasomes |
Linia Neutrogena Renutriv range, resilience range |
Johnson& Johnson Estee lauder |
|
Nano silicon dioxide |
lift makeup |
Lancome |
- Authors should explain why are nanomaterials make a special issue both cosmetic and medicine in safety and health consideration
Pollution of individual components of the environment with nanoparticles can have point character (i.e. plants producing nanoparticles or nanoproducts, waste incineration plants, waste landfills or sewage treatment plants) or surface (area). Most surface contaminants are associated with the release of nanoparticles during use. In addition to the unintentional release of NP into the environment, we also have the problem of deliberate introduction. An example is injected directly into groundwater contaminated with nZVI chlorinated solvents. In the environment, the formation of aggregates and therefore of larger particles that are trapped or eliminated through sedimentation affects the concentrations of free nanoparticles (Fig.3) [6]. Nanoparticles may pose an ecotoxicological risk in natural receivers and their bioaccumulation in the natural environment and potential inclusion in food chains may also affect human health [4, 23, 33]. People can either be directly influenced by NPs by exposure to air, soil or water or indirectly by consuming accumulated plants or animal NPs. Aggregated or adsorbed NPs will be less mobile, but collection by sedimented animals creates a risk of inclusion in the food chain [6].
According to the reports of the Scientific Committee for Emerging and Newly Identified Health Risks (SCENIHR), European Agency for Safety and Health at Work [66], not all nanomaterials are toxic. In order to understand well how nanomaterials work in a new product placed on the market, a case by case approach should be taken. As a result of the conducted research, the most important influence of nanomaterials was found in the lungs, including inflammation and tissue damage, pulmonary fibrosis and cancer. Nanomaterials can also have an effect on the circulatory system. Some types of carbon nanotubes can even have similar effects to asbestos. As with the lungs, nanomaterials can attack other organs and tissues, including the liver, kidneys, heart, brain, skeleton, and soft tissues.
- Authors should explanation required about perspectives on FDA’s regulation of nanotechnology in cosmetics and medicines
Products in the form of drugs and cosmetics easily get into the environment. In order to protect the environment against the ingress of nano-scale active materials, the level of nanomaterials in products, especially those newly introduced to the market, should be controlled. The U.S. plays an important role in the safety assessment and monitoring of the performance of cosmetics and drugs through continuous product reviews FDA (Food and Drug Administration) [80]. For this purpose, the Agency has developed a special research program aimed at obtaining as many tools and methods as possible to identify the properties of nanomaterials and their impact on products. The Agency takes a conservative scientific approach to assessing each product for its merits and does not make broad, general assumptions about the safety of nanotechnology products. In 2006, the FDA's nanotechnology working group assessed and identified possible knowledge or policy gaps to enable the agency to better assess the safety aspects of FDA regulated products. In 2007, the Nanotechnology Working Group published a report, which indicated recommendations for actions that the agency may take in implementing its mission of protecting and promoting public health [81]. With the increasing number of submissions of products containing nanomaterials, the Task Force strongly encourages internal research grants, provides rapid training in nanotechnology, encourages active participation in the development of international nanotechnology standards. FDA supports the National Nanotechnology Initiative (NNI) and collaborates with other agencies through participation in the Nanoscale Science Engineering and Technology (NSET) subcommittee and the Nanotechnology Environmental and Health Implications (NEHI) working group.
Since the publication of this report, the FDA has issued several guidance documents on topics related to the use of nanotechnology in FDA regulated products [80]. The guidance in the guidance does not create or confer any rights, it represents the current views of the FDA. In its 2014 reports, the Agency does not categorically evaluate nanotechnology, so it does not indicate that nanotechnology is inherently safe or harmful. Guidance takes into account the specific characteristics and effects of nanomaterials in the specific biological context of each product and its intended use. Specific approaches for each product area with nanomaterials (cosmetics [82], drugs, food products [83]) differ in scope and issues, for example, interactions of nanomaterials with natural systems, research approaches, product safety assessments, product quality, etc. from the Guidance for Industry Safety of Nanomaterials in Cosmetic Products [84], safety is determined by an extensive assessment of physical and chemical properties and by assessing contaminants, if any. The scope and impact of possible toxicity of nanomaterials is described by indicating the ways of exposure, absorption and penetration into organisms. Additionally, as indicated in the Guidance [84], toxicological studies in vito and in vivo, clinical studies, and toxicokinetics and toxicodynamics should be considered. Each cosmetic product should therefore contain a whole package of data and information justifying the safety of the product in terms of its conditions of use.
- Authors should public perception of nanotechnology and confidence in FDA
FDA's New Drug Application (NDA), Investigation New Drug (IND), Center for Devices and Radiological Health (CDRH), Center for Veterinary Medicine (CVM) controls and assesses the safety of nanomaterials based on the characteristics of the nanomaterial itself and toxicological analyzes [82, 84]. The FDA takes into account very broadly the impact of nanomaterials and the areas of their use. In the case of cosmetic, medical products that are to be placed on the market and have new or changed properties, the need for safety testing needs to be assessed. For food products, the assessment is made on a case-by-case basis. The FDA recommends a safety assessment taking into account e.g. physicochemical properties of nanomaterials, agglomerations and size distribution of nanoparticles, the presence and impact of possible pollutants, potential routes of exposure to nanomaterials, in terms of toxicity, it is recommended to conduct tests to obtain in vitro and in vivo toxicological data of nanomaterial components and their impurities, skin penetration, potential inhalation , irritation (skin and eyes) and sensitization tests, mutagenicity/genotoxicity tests.
- Authors should brief about existing gap between basic research relating nanomaterials and their application in cosmetics and medicine
Medicines are subject to stringent FDA-imposed controls for approval, but there are no such requirements for cosmetics. Cosmeceuticals are products on the border of cosmetics and pharmaceuticals. As of today, there is no rigorous control over the approval and regulation of nanocosmetics. No clinical trials are required for their approval, raising concerns about toxicity after use. Many cosmeceuticals change the physiological processes in the skin, but manufacturers still avoid clinical trials or making specific claims to avoid subjecting their products to costly and lengthy FDA approval processes. If the FDA determines that there is a problem with the safety of any cosmetic or ingredient, the FDA has the power to prohibit the sale and manufacture of the product [85] In the European Union, cosmetics are subject to the provisions of the Cosmetics Directive 76/768 / EEC. The EU doesn't have a category to call cosmeceuticals, but it does have strict rules where any company claims must be presented as evidence. According to the new EU regulation, manufacturers have to replace the nanoparticles contained in the product. Cosmetic regulations state that any product containing nanomaterials as an ingredient should be clearly listed and must insert the word "nano" in the brackets after the list of ingredients [86].
The gap between basic research concerns the impact of nanomaterials that got into the environment on plants, animals and, consequently, on human health. The phytotoxicity of nanomaterials is currently a topic that is not fully understood and researched. Most of the research to date has focused on germination, cell culture and genetic effects. There are studies showing an increase in the level of reactive oxygen species (ROS) in cells of higher plants under the influence of contact with nanoparticles, which, depending on the dose, resulted in cell death.
Assessing the toxicity and safety of nanoparticles requires an understanding of their uptake by organisms. Most studies focus on determining the nature of the phytotoxicity of nanoparticles, but quantitative methods for measuring them in plant or animal tissues have not been established. Nanoparticles larger than the pore size of the cell wall stick to the cells of the root epithelium, causing physical damage, clogging the pores and reducing the absorption of water and nutrients [90]. Through the pores in the cell walls, nanoparticles can easily penetrate. It is about their accumulation in plant tissues. As plants are an important food source for humans, further research is needed to evaluate the toxicity caused by nanomaterials. The mechanisms of metal-induced carcinogenesis are not well understood. Both genetic and non-genetic factors induced by nanoparticles in cells may predispose to carcinogenicity [91]. It is imperative to conduct research into the toxicity and genotoxicity of nanoparticles to be able to safely take advantage of the enormous potential benefits of this new technology [92].

Round 2
Reviewer 3 Report
Accept in present form